# Denoising Method for Passive Photon Counting Images Based on Block-Matching 3D Filter and Non-Subsampled Contourlet Transform

**DOI:** 10.3390/s19112462

**Published:** 2019-05-29

**Authors:** Xuan Wang, Liju Yin, Mingliang Gao, Zhenzhou Wang, Jin Shen, Guofeng Zou

**Affiliations:** School of Electrical and Electronic Engineering, Shandong University of Technology, Zibo 255000, Shandong, China; 18753370762@163.com (X.W.); sdut_mlgao@163.com (M.G.); wangzz@sdut.edu.cn (Z.W.); shenjin@sdut.edu.cn (J.S.); zgf841122@163.com (G.Z.)

**Keywords:** imaging system, multi-pixel photon counting detector, NSCT, block-matching 3D filter, SSR, image denoising

## Abstract

Multi-pixel photon counting detectors can produce images in low-light environments based on passive photon counting technology. However, the resulting images suffer from problems such as low contrast, low brightness, and some unknown noise distribution. To achieve a better visual effect, this paper describes a denoising and enhancement method based on a block-matching 3D filter and a non-subsampled contourlet transform (NSCT). First, the NSCT was applied to the original image and histogram-equalized image to obtain the sub-band low- and high-frequency coefficients. Regional energy and scale correlation rules were used to determine the respective coefficients. Adaptive single-scale retinex enhancement was applied to the low-frequency components to improve the image quality. The high-frequency sub-bands whose line features were best preserved were selected and processed using a symbol function and the Bayes-shrink threshold. After applying the inverse transform, the fused photon counting image was subjected to an improved block-matching 3D filter, significantly reducing the operation time. The final result from the proposed method was superior to those of comparative methods in terms of several objective evaluation indices and exhibited good visual effects and details from the objective impression.

## 1. Introduction

Dark environments often have a small amount of natural light, such as moonlight, starlight, and atmospheric glow. Collectively, this can be referred to as a low-light-level (3L) environment. Images taken in 3L environments generally have insufficient exposure and uneven illumination, resulting in deficiencies in the image detail information and a blurry imaging quality, which affects information extraction, analysis, and processing [1].

This study focuses on photon counting imaging, which is one approach used in 3L environments. In recent years, photon counting detectors with high detection efficiency and quantum efficiency have become a topic of interest in the field of 3L imaging [2]. Such detectors are designed to measure extremely weak optical signals and can detect targets under illumination levels of 10^−3^–10^−6^ lux (night-time illuminance values from 22:00–02:00 the next day, without fog and external artificial light sources, typically range from 0.001–0.3). The high detection efficiency and signal-to-noise ratio (SNR) of photon counting have led to its use in laser radar, quantum optics, bio-medicine, environmental radiation measurements, particle physics, high-energy physics, astronomy, and fluorescence measurements [3,4,5,6,7,8,9].

The multi-pixel photon counting detector (MPPC) used in this paper is essentially a silicon photo-multiplier (Si-PM), which is composed of a series of avalanche photodiode (APD) detectors operating in Geiger-counting mode [10]. Compared with common photon counting detectors such as a photo-multiplier tube (PMT), positron emission tomography (PET), and charge-coupled device (CCD), MPPC has better characteristics in terms of small size, low cost, higher reliability, and higher detection sensitivity [11]. The target is detected based on quantum and statistical theory, with the signal returned from the target regarded as discrete photons to be recorded. Analysis of the statistical properties of the returned photons generates the target photon values matrix, and the normalized version of this matrix is the grayscale image [12].

According to the different production mechanism, APD detectors generate various types of noise [13], such as shot noise, thermal noise, generation–recombination (g–r) noise, 1/f noise, avalanche excess noise, etc. [13,14]. Among them, thermal noise is similar to the shot noise, it is independent of frequency and belongs to white noise, which is generated by the random motion of carriers, and cannot be eliminated thoroughly. The 1/f and g–r noise have a high proportion of all noise and little influence on image quality improvement, and without any evident distribution characteristics, usually they are not considered when researching the denoising question. When the APD detectors of the MPPC are running, they are in the status of reverse bias, (the so-called Geiger counting mode), the avalanche excess and thermal noise are the main types of noise [11,15]. Avalanche excess noise is caused by the quantum effect during MPPC operation, which approximately obeys the Poisson distribution. The magnitude of it is affected by the impact ionization coefficient of electrons and holes, and is also related to the position of the photo-generated and thermal-generated current. In this paper, the avalanche excess and thermal noises are the main noises targeted and researched.

The goal of image denoising is to reconstruct a reasonable estimate of the original signal from the noise signal. Noise pixels should be removed while preserving features such as edges or texture details [16]. A regularized expectation-maximization (EM) algorithm has been developed for the maximum likelihood estimation of rate parameters in observed data [17], and this is combined with a scale-recursive Markov tree model for photon counting imaging. Experimental intensity estimation results on photon-limited images with real shot noise demonstrate the effectiveness of the proposed approach. Huajun et al. [18] proposed a photon counting denoising method based on the cross-validation theory of wavelet filtering, which significantly improves the SNR. A deep neural network has also been used to recover objects from the photon counting image, which demonstrates better performance for equivalent SNR [19]. Aiming at other 3L imaging methods such as underwater and medical imaging, Deshan et al. [20] applied a novel variational approach to underwater sonar images; the wavelet transform and gradient have been used to realize underwater range-gated image denoising [21]; and a model based on total variation (TV) and the split Bregman method have been applied to sea surface images at night [22]. In other imaging fields, convolutional neural networks, non-local means [23], and principal component analysis [24] have been applied to computed tomography or positron emission tomography medical imaging. The method based on principal component analysis and denoised block aggregation [24] even has a better denoising effect and more extensive usage than the block-matching 3D filter algorithm without selecting parameters by users.

Typical image denoising methods operate in either the spatial domain or the frequency domain. To overcome the limitations of wavelet transforms, such as the non-sparseness of high-dimensional time coefficients and the lack of multi-directional selectivity, multi-scale geometric analysis methods have emerged, including the contourlet transform (CT) [25] and ridgelet transform (RT) [26]. However, the conventional CT lacks translation invariance, and so the non-subsampled CT (NSCT) was proposed by Cunha et al. [27] to avoid the down-sampling process. A Gaussian distribution model has been applied to the NSCT domain to obtain a reasonable denoising effect [28], but the model cannot remove noise that is concentrated in the highest scale sub-band. The non-Gaussian distribution model [29] has been used to model the NSCT coefficients as well as the neighborhood coefficients and parent coefficients. The classification criterion is divided into two parts according to the degree of importance. Although the subjective effect does not provide sufficient improvement, the classification criterion for the correlation between coefficients is worth learning and improving.

Focusing on these problems, this article describes an improved denoising framework for photon counting images. The remainder of this paper is organized as follows. Section 2 introduces the overall framework and explains the design of each component. Section 3 expands on the basic theory behind each part of the framework and the improvement offered by our method, as experimental platform introductions and image simulation and analysis are presented in Section 4. Finally, Section 5 summarizes our conclusions from this study.

## 2. Overall Framework

The overall framework described in this paper is shown in Figure 1. The source photon counting image and histogram-equalized image are transformed to the NSCT domain and divided into low- and high-frequency coefficients. Different fusion rules are used according to their respective characteristics. At the same time, adaptive single-scale retinex (SSR) and a threshold function are used to process the coefficients. Finally, the NSCT image is denoised using an improved block-matching 3D filter (BM3D).

Under 3L environments, the photon counting image contains lots of noise, the brightness and contrast of the image are extremely low, and details from the source image cannot be seen. At present, in common enhancement algorithms, histogram equalization and its improved algorithms suffer from excessive equalization, especially in dealing with 3L images. After 0–1 normalization of the photon counting image, most pixel values are concentrated from 0–0.2, and equalization significantly reduces the level of grayscale, extends the quantized interval, and blurs the image. Enhancement algorithms based on retinex exhibit good results in terms of the enhancement, restoration, and dehazing of RGB images. For grayscale photon counting images, excessive whitening of the image may occur, possibly because the algorithm perceives a pixel point depending not only on the absolute light entering the human eye, but also on the color and brightness around the pixel point. For grayscale images, the photon counting image has less color discrimination and extremely low brightness, so the algorithm performs poorly.

Considering the above discussion, the first improvement is to apply NSCT to both the photon counting image and the histogram-equalized image simultaneously. According to the characteristics of the high- and low-frequency components, two fusion rules are then applied to the corresponding region whose distribution of detail information and noise are different. The fusion of a high-brightness image and a low-brightness image can prevent extreme illumination values from occurring.

The second improvement is the use of a fusion rule based on the region energy and adaptive SSR algorithm for the low-frequency sub-band. This enhances the brightness and corresponds to the use of SSR, but has a lower computational complexity and is suitable for processing grayscale images. As for the high-frequency sub-bands, although most noise points are in the high-frequency domain, only a fraction of the detail information is included in this region. The details are mainly concentrated in the first high-frequency layer, which has the best line features. Thus, the coefficients of high-frequency sub-bands are fused by considering the correlation between the different values at the same scale, and other coefficients are proposed using a symbol threshold function. Finally, a block-matching 3D filter is used to eliminate the noise points. Because the image has been processed once in the previous NSCT section, it is not necessary to search repeatedly, so we simplify the process of grouping by block-matching. This process speeds up the operation, but does not affect the noise reduction performance.

## 3. Methodology

### 3.1. NSCT

As the base function of the two-dimensional wavelet is isotropic, the directional choice is often poor. This lack of directionality means that the two-dimensional wavelet transform is unable to take full advantage of the geometric regularity of the image itself, and cannot obtain the best sparse representation [29,30] of the line or plane singularities in the image. Contourlet transform offers multi-directionality and anisotropy [25], and so fewer coefficients are needed to sparsely represent the image in an optimal way and efficiently capture curved and oriented geometrical structures in the image. However, because CT lacks translation invariance, NSCT is the better choice, as it discards the down-sampling operation in CT and combines the non-subsampled pyramid (NSP) and non-subsampled directional filter banks (NSDFBs) [31]. After the transformation, the sub-bands in every direction on each scale have the same size as in the source image. Figure 2 illustrates the whole process of transformation for the example of a two-layer structure [30].

### 3.2. Determining Sub-Band Coefficients

#### 3.2.1. High-Frequency Sub-Band Coefficients

Because this study focuses on the denoising and improvement of visual effects, we only investigate some common fusion rules to integrate the high- and low-frequency segments. Additionally, we do not provide any in-depth discussion of the influence of different fusion rules on the coefficients. According to the correlation among the high-frequency coefficients in the same scale but different directions, Equation (1) is used to select coefficients with relatively large values [32].
(1)$${\hat{W}}_{k,l}\left(i,j\right)=\{\begin{array}{cc} A_{k,l}\left(i,j\right) & \left|\sum_{l=1}^{L}A_{k}\left(i,j\right)\right|>\left|\sum_{l=1}^{L}B_{k}\left(i,j\right)\right|\\ B_{k,l}\left(i,j\right) & otherwise \end{array}$$


The judgment condition is the summation of the L-layer sub-band coefficients in the same scale, so it connects each decomposition level and direction to determine the fused coefficient (A and B represent the high-frequency coefficients from the photon counting image and the histogram-equalized image, respectively). As the decomposition scale and the number of directions increase, the sub-bands contain less detail. The overly complex process for high-frequency sub-bands makes no significant improvement effect, but obviously increases the computation time. Thus, we only apply the fusion process to the first-layer coefficients, and propose a new symbol function and use a Bayes-shrink threshold for the other sub-bands.

The new symbol threshold function designed in this study is expressed in Equations (2) and (3).
(2)$${\hat{W}}_{k,l}\left(i,j\right)=\{\begin{array}{ccc} uW_{k,l}\left(i,j\right)+(1-u)\mathrm{sgn}(W_{k,l}\left(i,j\right))(\left|W_{k,l}\left(i,j\right)\right|-\lambda_{1}/\left(\exp{\left(\left|W_{k,l}\left(i,j\right)\right|-\lambda \right)}^{k}\right)) & \left|W_{k,l}\left(i,j\right)\right|\geq \lambda & \\ \left|W_{k,l}\left(i,j\right)\right|-\mathrm{sgn}(W_{k,l}\left(i,j\right))\times \lambda_{1} & \lambda_{1}\leq \left|W_{k,l}\left(i,j\right)\right|<\lambda & \\ 0 & \left|W_{k,l}\left(i,j\right)\right|<\lambda_{1} & \end{array}$$
(3)$$u=1-\exp^{-1\times m\times {\left(\left|W_{k,l}\left(i,j\right)\right|-\lambda \right)}^{}}$$
where λ and $\lambda_{1}$ are dual threshold and $\lambda_{1}=\varepsilon \lambda$, with ε∈(0,1) (ε=0.3 in the study). k represents the scale and m is a constant parameter. $W_{k,l}\left(i,j\right)$ represents the original fusion coefficient. The function is continuous at the threshold of ±λ because of the symmetry about the origin; if $W_{k,l}\left(i,j\right)\to \lambda^{+}$, u→0
(4)$$\underset{W_{k,l}\left(i,j\right)\to \lambda^{+}}{\lim}{\hat{W}}_{k,l}\left(i,j\right)=\lambda -\lambda_{1}=\underset{W_{k,l}\left(i,j\right)\to \lambda^{+}}{\lim}{\hat{W}}_{k,l}\left(i,j\right)$$
the function is continuous at the threshold of ±λ. Similarly, the second threshold $\pm \lambda_{1}$ has the same property and is continuous at the point of discontinuity. As for the deviation, when $W_{k,l}\left(i,j\right)\to +\infty$, u→1
(5)$$\underset{W_{k,l}\left(i,j\right)\to +\infty }{\lim}\left[{\hat{W}}_{k,l}\left(i,j\right)-W_{k,l}\left(i,j\right)\right]=\underset{W_{k,l}\left(i,j\right)\to -\infty }{\lim}\left[{\hat{W}}_{k,l}\left(i,j\right)-W_{k,l}\left(i,j\right)\right]=0$$

The function takes into account the advantages of the soft and hard threshold functions, and solves the problem of discontinuities and fixed deviations. Thus, the dual threshold setting can be finely adjusted to the data.

#### 3.2.2. Low-Frequency Sub-Band Coefficients

The low-frequency sub-bands include most of the energy of the source image, effectively reflecting the background information. Therefore, it may not be possible to obtain significant salient features, even though a high-pass constraint is imposed. A simple fusion rule is the weighted mean:
(6)$${\hat{W}}_{k,l}\left(i,j\right)=w_{1}A_{k,l}\left(i,j\right)+w_{2}B_{k,l}\left(i,j\right)$$
(7)$$w_{1}+w_{2}=1$$
where $w_{1}$ and $w_{2}$ are the weights. However, this method can lose some approximate information, reduces the level of the grayscale, and extends the quantized intervals. Thus, we use a fusion rule based on the region energy for the low-frequency coefficients. First, we compute
(8)$$e_{{}_{k}}^{l}=\sum_{i=1}^{M}\sum_{l=1}^{N}W_{k,l}{}^{2}\left(i,j\right)\;\mathrm{and}\;\eta =e_{{}_{k}}^{l}/\sum e_{{}_{k}}^{l}$$
where $e_{{}_{k}}^{l}$ represents the energy of $W_{k,l}\left(i,j\right)$, the selected region contains M × N pixels and has its energy center at (i,j), η is the weight ratio coefficient, and $\sum e_{{}_{k}}^{l}$ represents the total energy in the specified region. The fusion rule is:(9)$${\hat{W}}_{k,l}\left(i,j\right)=\{\begin{array}{cc} A_{k,l}\left(i,j\right) & \eta_{A(i,j)}>\eta_{B(i,j)}\\ B_{k,l}\left(i,j\right) & otherwise \end{array}$$
we can determine the choice of coefficients ${\hat{W}}_{k,l}\left(i,j\right)$ based on the magnitude of the comparison according to $\eta_{A(i,j)}$ and $\eta_{B(i,j)}$. Next, we use adaptive SSR enhancement in a process of continual optimization. Retinex theory [33,34,35] is based on how the human visual system adjusts the perceived object color and brightness model. The basic assumption is to consider an image I(x,y) to be formed from the illumination component L(x,y) and reflectance component R(x,y) as:
(10)I(x,y)=L(x,y)×R(x,y)

The core concept of retinex is to estimate the illumination, and many improved versions have been proposed, such as the central/surround algorithm [36], SSR [37], and MSR [38]. The use of scale factors and weighted normalization after filtering can achieve a balance among the gray dynamic range compression and edge enhancement. Considering its applicability and computational complexity, SSR is the best choice.

We use a Gaussian low-pass filter as the surround function for estimating the light component:
(11)$$F\left(x,y\right)=\frac{1}{\sqrt{2\pi }\sigma^{2}}\times \exp\left\{-\left(x^{2}+y^{2}\right)/2\sigma^{2}\right\}$$
where ∬F(x,y)dxdy=1 and σ is a fixed parameter. We use an adaptive scale parameter $\sigma_{1}$, which is given by the local adaptive standard deviation:
(12)$$\sigma_{1}=W_{}\left(i,j\right)+\frac{\sigma_{w}-\sigma_{\min}}{\sigma_{\max}-\sigma_{\min}}\times C_{mean-scale}$$
where $\sigma_{w}$ represents the standard deviation of a set window (such as 3 × 3 or 5 × 5 pixels), and $\sigma_{\max}$ and $\sigma_{\min}$ represent the maximum and minimum standard deviation of all windows. W(i,j) is the gray mean value in the selective region, and $C_{mean-scale}$ is the value level of the grayscale level in the corresponding window region.

### 3.3. Block-Matching 3D Filter

The block-matching 3D filter [39,40] is based on the non-local mean. The basic algorithm includes a basic estimate and a final estimate. Each estimate can be divided into block-matching, associated filtering, and aggregation. The primary block diagram is shown in Figure 3.

The image is divided into many blocks of a given size according to the similarity among blocks, and two-dimensional images with similar structures are combined to form a three-dimensional array from which noise can be separated using some associated filtering method, such as the threshold process or the Wiener filter. The associated filtering process ensures that the matched image similarity blocks are transformed into the spectral domain, and weakens the noise by decreasing the corresponding coefficients. The estimated value of each image block can be obtained by applying an inverse transform. The processed image estimation values are then aggregated, and the image blocks whose corresponding locations overlap are weighted and averaged to obtain the final result [41].

The main improvement is the process of grouping by block-matching. An *N* × *N* window moves across the image according to the set step-size, forming lots of image blocks. The similarity among the image blocks is measured by some metric such as the Euclidean distance:(13)$$l_{p}={\left\|p_{x}-p_{y}\right\|}_{2}^{2}/\left(N\times N\right)$$
for image p, ($p_{x}$, $p_{y}$) represents the image block whose top-left corner is located at pixel point (x,y) and ${\left\|p_{x}-p_{y}\right\|}_{2}^{2}$ represents the Euclidean distance between two image blocks. After choosing a reference block, $l_{p}$ is used to compare the similarity of the reference image block to image blocks in different positions to realize grouping. Because the reference block can be specified arbitrarily, a global calculation across all image blocks would be computationally intensive. To reduce the search space and accelerate the calculation, we use *h* similar blocks that have the smallest distance from the reference block to form a three-dimensional matrix. This not only reduces the number of searches, but also approximately guarantees the same results as a global search. If the picture has size M×M and the step size is c, $M^{2}$ matching calculations are required for a global search, compared with just ${\left(\frac{M}{c}\right)}^{2}+h{\left(c-1\right)}^{2}$ for our modified search. This is a significant reduction in the computational expense.

## 4. Experimental Introduction

### 4.1. Experimental Setting

The two-dimensional optical experiments conducted in this study used a platform composed of a multi-pixel photon counting detector, SC stepper motor controller, electronic control rail, LSW101 tungsten light source, wide-range micro illuminometer, computer, optical fiber, and a cable. A diagram of this experimental platform and an image of the real experimental platform are shown in Figure 4 and Figure 5, respectively.

The process of obtaining the two-dimensional photon counting image is as follows. First, the stepper motor controls the two-dimensional rail to scan the target image line by line. Next, the reflected photon values of the target image, which are determined according to the Poisson distribution, are transmitted to the multi-pixel photon counting detector through the optical fiber. Finally, the data are counted by a software program in the computer. After reconstruction and 0–255 normalization of the photon counting values, the final values represent the pixels of the grayscale photon counting image.

All experimental results were obtained on an Intel(R) Core (TM) i7-8750H with 16 GB RAM using MATLAB 2016b. We got two photon counting images using our platform and tested the denoising effect. The test images had resolutions of 120 × 120 and 100 × 100 pixels, respectively. The multi-pixel photon counting detector used in this experiment detected the pixel value of the target image by dot matrix scanning. The higher-resolution image required a longer scanning time, and other factors such as uneven illumination influenced the imaging quality. The parameter settings of NSCT are given in Table 1.

The “maxflat” is a multiscale filter and “dmaxflat” is a directional filter, they are the built-in function in MATLAB. In the process of setting the NSCT parameters, we found that more decomposition layers (>3 layers) and direction numbers (>16) did not improve the visual effect enough to justify the enhanced operation time. Thus, we selected the specific values in Table 2. If the photon counting image was to be denoised directly, the low brightness and contrast mean that no obvious denoising effect would be observed, regardless of which algorithm was used. The results would look no different to those in Figure 6. Therefore, it is necessary to combine image enhancement to achieve the associated processing effect.

The performance of the proposed method was compared with that of the following classical methods: wavelet threshold (WT, wavelet basis: “db1”, decomposition layer: “5”, proposed by the hard threshold function and Bayes-shrink threshold), wavelet combined with fast independent component analysis (WA-FICA, number of iterations: “100”, convergence threshold: “10^−6^”), non-subsampled contourlet transform (NSCT, parameters’ setting can refer to Table 2), and block-matching 3D filter (BM3D, parameters’ setting can refer the source code in literature [39]). When comparing the new method with the above listed algorithms, the image was enhanced by histogram equalization (HE) before denoising to ensure a relatively fair comparison between pictures under same levels of illumination. The experimental results are shown in Figure 7 and Figure 8. Then, we analyzed the image denoising effect from subjective visual effect and objective indices evaluation.

### 4.2. Experiment Simulation and Analysis

In Figure 7, the red and blue boxes mark areas of the target from which we can judge the clarity of the details and the denoising effect. Figure 7c,d exhibit obvious noise points across the image, and the outline of the cups in the red and blue boxes is blurred. The outline of the cup in the blue box can be seen, but the pixel values of the cup in the red box are blurred and do not contain any useful information. Comparing Figure 7e with Figure 7c,d, the noise points are largely eliminated, but the improvement in details is not significant. In Figure 7f, all of the noise points have been eliminated, but the picture looks too smooth to be natural. The biggest change in Figure 7g is that, in addition to the above advantages of other algorithms, the outline of the cup in the red and blue boxes is very clear, especially at the handle, and the cup holder can be clearly seen. This shows that the algorithm proposed in this paper achieves a balanced improvement between preserving the details and denoising, a result of the fusion of NSCT and the BM3D filter.

The image in Figure 8a was detected under a 4.17 × 10^−4^ lux environment, an order of magnitude lower than the brightness for Figure 7a. Hence, denoising is theoretically more difficult and the noise points of the histogram-equalized image are more notable (see Figure 8b). The blue rectangular box around the face of Lena can be used to compare the details after denoising. Figure 8c,d is again unsatisfactory, containing lots of noise blocks rather than the noise points in Figure 8b without improving the presentation of details. In Figure 8e, the image processed by NSCT eliminates noise points, but is too smooth to preserve the edge details, suggesting a poor balance between preserving details and denoising. Figure 8f displays a good denoising effect, but has the same problem as Figure 8e in that it is too smooth to preserve details. In Figure 8g, the balance of detail retention and noise removal is better. Details such as the messenger wire, eyes, hat, and mouth are more distinct and cleaner within the blue area. Thus, our method can greatly suppress noise and retain edge details of the face and hat under an illumination environment of 10^−3^–10^−4^ lux, whereas the other methods cannot.

Next, we used some blind image quality assessment (IQA) indices and information entropy (IE) to evaluate the image quality. The reason we did not use the full reference IQA indices such as SSIM, PSNR, and MSE was that we did not have an undistorted image as reference and to contrast, so the results were meaningless. We researched published blind IQA indices such as GQNR [42], NIQE [43], and BRISQUE [44]. They can give blind images an objective score without any reference information. The BRISQUE algorithm is operated in the spatial domain and this method does not compute specific distortion but instead uses scene statistics of locally normalized luminance coefficients to quantify possible losses of “naturalness” in the image due to the presence of distortions, thereby leading to a holistic measure of quality, and it has very low computational complexity, and is suitable for real time applications. The greatest advantages of NIQE is that it only makes use of measurable deviations from statistical regularities observed in natural images, without training on human-rated distorted images. The results are more objective and fairer for evaluating the denoising quality. The GQNR is the first proposed method that can blindly assess pan-sharpened image quality where the pan band does not overlap with the image bands. From the experiment result, it indicates that their method has better performance because the new objective IQA tool without reference for fused images was used as the basis of comparison, but this paper does not offer codes which can be used directly for testing, so we used NIQE and BRISQUE as testing indices.

Information entropy (IE) is a statistical form of a feature that reflects the amount of average information in an image. The formula is shown in Equation (13).
(14)$$H=-\sum_{i=0}^{255}P_{i}\log P_{i}$$
where *P_i_* is the probability of a grayscale value appearing in the image and can be obtained from the gray histogram. *H* represents the value of IE, the larger value of it, the more information in an image and it is better too. The data results are presented in Table 2 and Table 3. A comparison of the various indices are shown in Figure 9.

From Figure 9a, the proposed method has the highest IE values, demonstrating that it includes more information. The lower score for NIQE and BRISQUE represents the better quality. From Figure 9b,c, we can find that compared with respective reference image, the new method in this paper has the best objective quality in the listed algorithms. Next is the block-matching 3D filter method, but it still has a performance gap compared with the new method in this paper. For some methods like WA, WA-FICA, and NSCT, their denoising quality are even worse than the reference image. In other words, the cost of denoising is to lose details and decrease the image’s quality. We can get similar conclusion by comparing the data of NIQE and BRISQUE in Figure 7. The proposed method is best, followed by BM3D, WA, WA-FICA, and NSCT, but the conclusions are different in Figure 8. Except for the new method and BM3D, there is no specific order for WA, WA-FICA, and NSCT from the data which we think is caused by bias in different evaluation criteria or errors.

The enhanced image using the proposed method in Figure 8g seems to have better quality than Figure 7g, even though Figure 8a is collected in a worse illumination conditions compared with Figure 7a. We think there are two reasons to explain this. First, Figure 8 was obtained by scanning the Lena image, which is a widely used standard digital image, which has better contrast and brightness level, and is suitable for verifying various algorithms. Second, the enhancement algorithms may need more adaptive parameters, and in Figure 7, the enhancement algorithms may have been too strong and overcompensated.

Our work has mostly focused on the illumination levels of 10^−3^–10^−4^ lux. The reason for this is that it is not necessary to use passive photon counting devices to detect targets in the illumination levels of 10^−1^–10^−2^ lux, because the CCD or other detectors already have good imaging quality with little noise information. If you need to collect and dispose of images in illumination levels of 10^−1^–10^−2^ lux, we recommend the common parameter settings referred to in Figure 7 or Figure 8—both of which have a good proposed effect. But in environments with lower illumination levels such as 10^−5^–10^−6^ lux, our method cannot obtain an evident denoising effect even though we used best performance parameters without considering operation speed. The prime reason for a worse effect, we suggest, is that the noise mechanism was changed and the noise distributions did not correspond to presupposition, and some small changes can have random effects on the quality of the image. However, we have not researched in-depth about this and our future research will focus on noise distributions in different illuminations and propose algorithms with wider applicability.

## 5. Conclusions

A denoising method for passive photon counting images based on a block-matching 3D filter and NSCT was proposed. Test results show that the proposed method achieved significant denoising and detail enhancement effects on images obtained under illumination levels of 10^−3^–10^−4^ lux. Objective evaluation indices show that our method provides relatively good performance. In general, the proposed approach is an effective method for this kind of passive photon counting image, but it does not have universal applicability for all 3L environments, especially at extremely low illumination levels. This is a defect in the proposed method and we will continue to research specific noise types and distribution under different illumination levels and better expand the algorithm’s applicability.

## Figures and Tables

**Figure 1 sensors-19-02462-f001:**
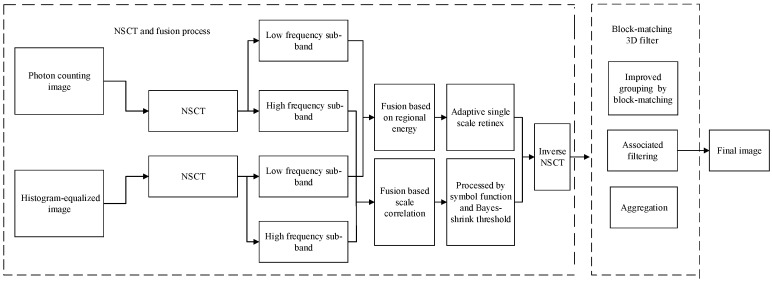
Overall framework presented in this paper.

**Figure 2 sensors-19-02462-f002:**
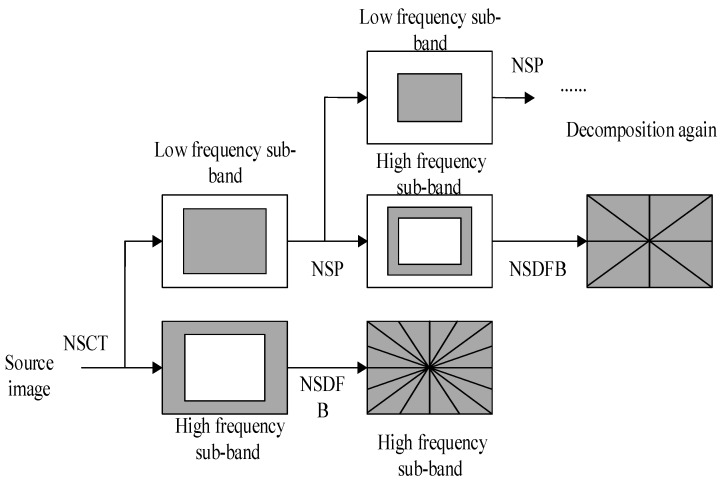
Decomposition of non-subsampled contourlet transform (NSCT).

**Figure 3 sensors-19-02462-f003:**
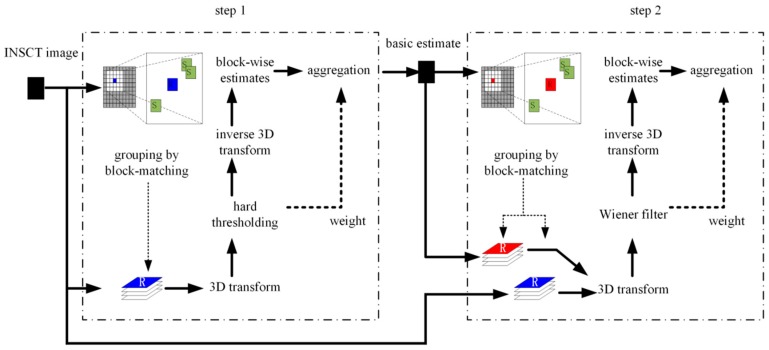
Principle of the blocking-matching 3D filter.

**Figure 4 sensors-19-02462-f004:**
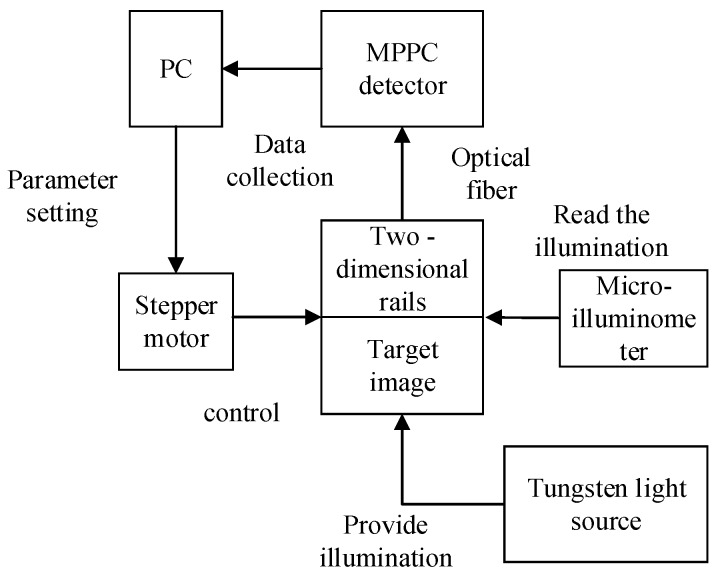
Principle diagram of optic experiment platform.

**Figure 5 sensors-19-02462-f005:**
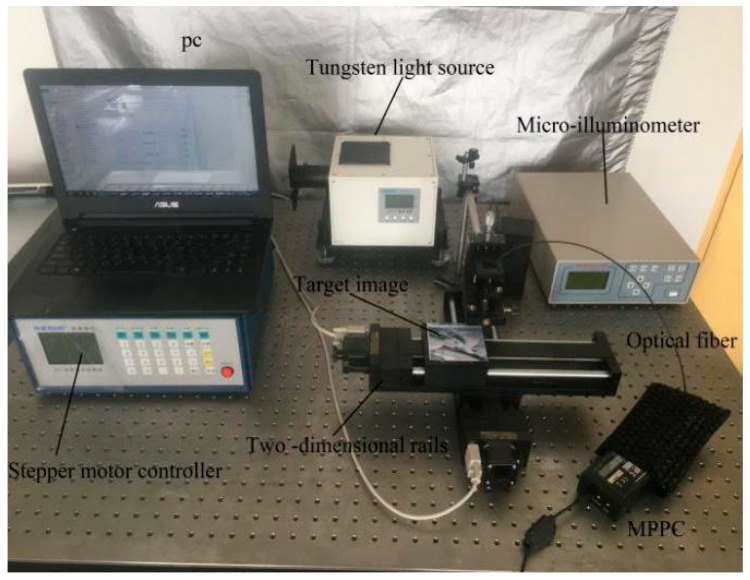
Image of real optic experiment platform.

**Figure 6 sensors-19-02462-f006:**
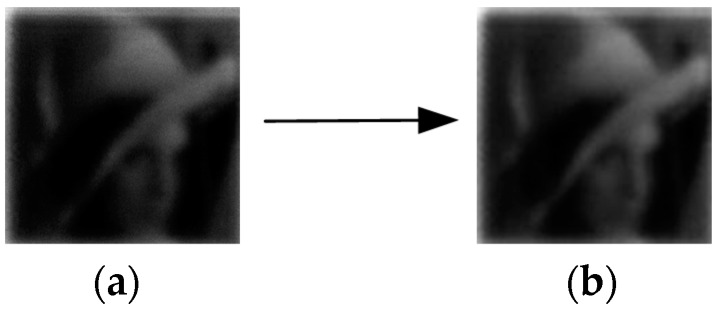
Denoising effect of photon counting image without any enhancement such as contrast and brightness: (**a**) photon counting image; (**b**) denoised image using block-matching 3D filter.

**Figure 7 sensors-19-02462-f007:**
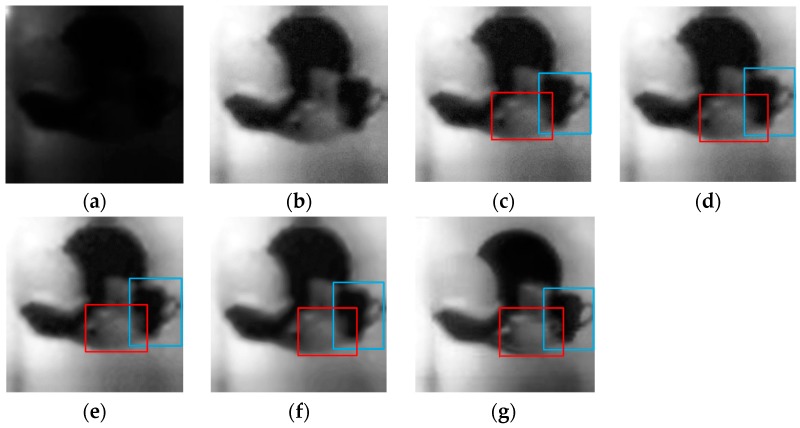
Comparison of different algorithms’ denoising effects on a photon counting image of a combination of balls and cups (detected under 2.19 × 10^−3^ lux environment with scanning resolution of 100 × 100). (**a**) Photon counting image; (**b**) reconstruction by HE; (**c**) reconstruction by WA-based HE; (**d**) reconstruction by WA-FICA-based HE; (**e**) reconstruction by NSCT-based HE; (**f**) reconstruction by BM3D -based HE; (**g**) reconstruction by proposed algorithm. The decomposition layer was three, direction number was 4, 8, and 16 in NSCT, the block size was 8, the sliding step to process every next reference block was 3, and the maximum number of similar blocks was 16. The threshold for the block distance was 3000 and the filtering parameters are referred to in the literature [39].

**Figure 8 sensors-19-02462-f008:**
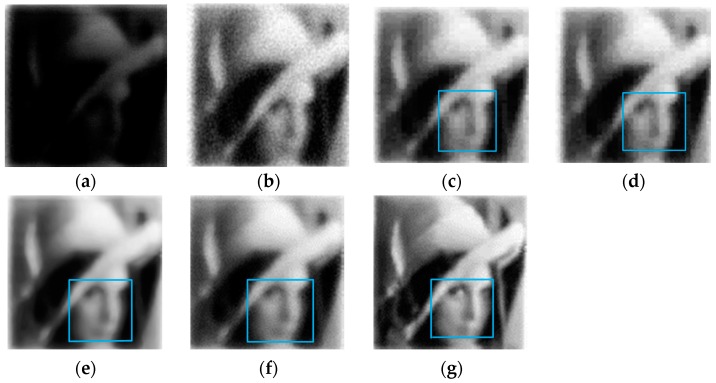
Comparison of different algorithms’ denoising effect on the Lena photon counting image (detected under 4.17 × 10^−4^ lux environment with a scan resolution of 120 × 120). (**a**) Photon counting image; (**b**) reconstruction by HE; (**c**) reconstruction by WA-based HE; (**d**) reconstruction by WA-FICA-based HE; (**e**) reconstruction by NSCT-based HE; (**f**) reconstruction by BM3D -based HE; (**g**) reconstruction by proposed algorithm. The decomposition layer was 4, the direction number was 4, 8, and 16 in NSCT and the block size was 6, the sliding step to process every next reference block was 3, the maximum number of similar blocks was 16, the threshold for the block-distance was 2000, and the filtering parameters are referred to in the literature [39].

**Figure 9 sensors-19-02462-f009:**
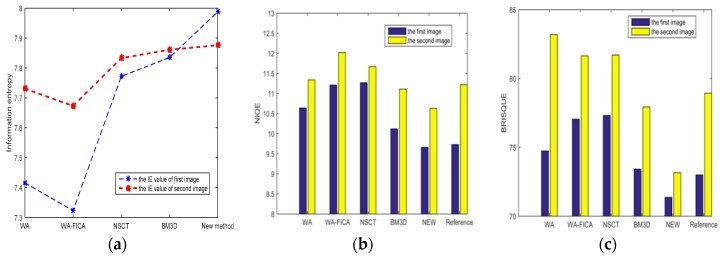
Comparison of different indexes: (**a**) IE; (**b**) NIQE; (**c**) BRISQUE.

**Table 1 sensors-19-02462-t001:** Parameter settings for NSCT.

Parameter	Setting
Tower decomposition	“maxflat”
Directional filter bank	“dmaxflat”
Decomposition layers	3
Choice of direction number in high-frequency Case	4, 8, 16

**Table 2 sensors-19-02462-t002:** Comparison of different algorithms’ denoising effects from Figure 7.

Photon Counting Image (Figure 7)	The Comparison of Different Algorithms				
Objective criteria	WA	WA-FICA	NSCT	BM3D	Proposed method
Information entropy	7.4156	7.3231	7.7722	7.8351	7.9878
NIQE (Figure 7b: 9.7321)	10.6395	11.2090	11.2686	10.1233	9.6569
BRISQUE (Figure 7b: 72.9922)	74.7423	77.0459	77.3150	73.4175	71.3675

**Table 3 sensors-19-02462-t003:** Comparison of different algorithms’ denoising effects from Figure 8.

Photon Counting Image (Figure 8)	The Comparison of Different Algorithms				
Objective criteria	WA	WA-FICA	NSCT	BM3D	Proposed method
Information entropy	7.7314	7.6729	7.8332	7.8609	7.8758
NIQE (Figure 8b: 11.2209)	11.3358	12.0218	11.6670	11.1077	10.6329
BRISQUE (Figure 8b: 78.9403)	83.2024	81.6526	81.7108	77.9424	73.1561

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
