# Peer review of "Denoising Method for Passive Photon Counting Images Based on Block-Matching 3D Filter and Non-Subsampled Contourlet Transform"

_sensors, 2019, doi:10.3390/s19112462_

Reviewer 1 Report

Please see the attached report.

Author Response

 Please find the responses in the attachment.

Reviewer 2 Report

I have some concern about some statements in the Abstract, for example, “Multi-pixel photon counting detectors can produce images in low-light environments using the photon counting principle. However, the resulting images suffer from problems such as low contrast, low brightness, and some unknown noise distribution.” What does “photon counting principle” mean? The problems mentioned here are usually for passive photon counting imaging. Make sure to clarify the topic of the manuscript is more related to passive photon counting imaging. Otherwise, some discussion in the abstract is just so confused. For example, for time-correlated single-photon counting based imaging LiDARs which are active imagers, it is possible to have high-quality intensity and depth images by taking advantage of their temporal filtering with synchronization between pulsed illumination and single-photon receiver.  Also, for typical (non low-light-level) passive optical imaging, the images could be degraded by some sort of unknown noise source, so it is not right only for photon counting imaging.  Please concentrate on discussions of the noise sources included (for example, Lines 64-65, thermal noise and avalanche excess noise)

Figure 3 is missed in the manuscript.

Line 67: what does “damaged signal” mean?

Figures 7 and 8, how did you calibrate the light levels (i.e. numerical values in terms of lux) of the environment to capture the images?

Does this denoising method is a joint approach of BM3D and NSCT?

Why is the method only available at a range of light level of the environment? Since this, are there specific applications in line with these? This point should be included in the abstract for the elevation of the method.

Author Response

Dear reviewer:

Aiming at questions your proposed for my manuscript, here are my explanations.

The question 1: I have some concern about some statements in the Abstract, for example, “Multi-pixel photon counting detectors can produce images in low-light environments using the photon counting principle. However, the resulting images suffer from problems such as low contrast, low brightness, and some unknown noise distribution.” What does “photon counting principle” mean? The problems mentioned here are usually for passive photon counting imaging. Make sure to clarify the topic of the manuscript is more related to passive photon counting imaging. Otherwise, some discussion in the abstract is just so confused. Please concentrate on discussions of the noise sources included (for example, Lines 64-65, thermal noise and avalanche excess noise)

1. The reply to first question.

Thanks for your advice, we have modified ”photon counting image” with ”passive photon counting image” all over the paper to distinguish the differences between the passive imaging and active imaging. The mean of ”photon counting principle” is referred to the process of getting the number of photons based on Poisson distribution. In order to avoid the ambiguity, we modified ”photon counting principle” with ”passive photon counting technology”. A simple instruction about it in the paper are the lines of 51-55.

In addition, we have re-discussed the noise sources in the lines of 56-69.

According to the different production mechanism, APD detectors generate various types of noise, such as shot noise, thermal noise, generation–recombination (g-r) noise, 1/f noise, avalanche excess noise and so on. Among them, thermal noise is similar to the shot noise, it is independent of frequency and belongs to white noise, which is generated by the random motion of carriers, and can not be eliminated thoroughly. The 1/f and g-r noise have a low proportion of all noise and have little influence on image quality improvement, and without any evident distribution characteristics, usually they are not considered when research denoising question. When the APD detectors in MPPC are running, they are in the status of reverse bias, (the so-called Geiger counting mode), the avalanche excess and thermal noise are main types of noise. Avalanche excess noise is caused by the quantum effect during MPPC operation, which approximately obeys the Poisson distribution, and its magnitude is affected by the impact ionization coefficient of electrons and holes. The magnitude of it is also related to the position of the photo-generated and thermal-generated current. The avalanche excess and thermal noise as the main research object,used in this paper.

The question 2: ”Figure 3 is missed in the manuscript and in line 67”: what does ”damaged signal” mean?

2. The reply to the second and third question.

We have added the Figure 3 in the manuscript. The original meaning of ”damaged signal” is the blind image with noises, we have modified it with ”noise signal” to prevent ambiguity.

The question 3: Figures 7 and 8, how did you calibrate the light levels (i.e. numerical values in terms of lux) of the environment to capture the images?

3. The reply to the second and third question.

We use Micro-illuminometer to detect the illuminance of the circumstance, the status of operating is showed in the Fig.1 and we can read the value of illuminance precisely. After each test, we would choose the operation of ”reset” to restore calibration and initialization. Due to limited experiment conditions, we can not set a specific value precisely, only reach the required illumination level as much as possible to collect the reflected photons.

          Fig 1: micro-illuminometer

The question 4: Does this denoising method is a joint approach of BM3D and NSCT?

4. The reply to the fourth question.

Yes, it is a joint approach of BM3D and NSCT. NSCT has good multiscale characteristics, block-matching 3D filter is used to eliminate the noise points. Because the image has been processed once in the previous NSCT section, it is not necessary to search repeatedly, so we simplify the process of grouping by block-matching. This speeds up the operation, but does not affect the noise reduction performance.

The question 5: Why is the method only available at a range of light level of the environment? Since this, are there specific applications in line with these? This point should be included in the abstract for the elevation of the method.

5. The reply to the fifth question.

We considered this problem carefully and think that this statement was very imprecise like ”only available at a specific light level of the environment” in the conclusion section. The explanation has showed in the line 413-422. we did not research the illumination levels of  lux because the enhancement and denoising effect has already did well using other detectors like CCD detector. But in the illumination levels such as  lux, our method can not get a evident denoising effect, the prime reason we guess is that the noise mechanism is changed and the noise distribution are not correspond to presupposition, some small changes can have random effects on the quality of the image. Through testing we find that the new algorithm has a good denoising effect on the the illumination levels of  lux.

As for the specific applications, In this extreme environment, this method is suitable for space exploration, bio-medicine imaging etc. This point has included in the abstract for the elevation of the method.

Reviewer 3 Report

Review Report – Sensors - 493424

• A brief summary  

This article tried to describe and propose new improved denoising framework for photon counting images, with additional qualities of speeding up the operations and calculations, i. e. significant reduction in the computational expense, without affecting the noise reduction performance.

• Broad comments  

Introducing overview was partially supported by methodology used to fulfil aim of the experiment. Experimental design is weak and it cannot be entirely supported by proposed methodology in order to bring stable results, and henceforth - objective conclusions. Choice to cover only two different images - each one in their sole one brightness magnitude – shrunk all relevance of positioning proposed algorithm among other standard ones within ‘objective criteria’. Therefore, limitations presented before and in conclusions, show awareness of authors on this major flaw (authors correctly recognized and narrowed most open issues to further research).

• Specific comments

1.    Line 220 - Figure 3. missing

2.    Equation from line 175 - put in separate line: equation (4) … equation (4) becomes (5) etc…

3.    References&authors often not related to text: e.g. ref 206 line 77, ref 33-36 with line 203 etc..

4.    Subjective judgement (mentioned in line 265, later in 265) must always be presented with and followed by construction and validation protocol

5.    Figures 7g) and 8 g) offer good representation of effects of proposed algorithm, mostly supported by values within objective criteria. However, presented example may seem highly selective or accidental, therefore concluding sentence in line 381 and 382: “that the proposed method achieves significant denoising and detail enhancement effects on photon counting images obtained under illumination levels of 10-3to 10-4 lux”, simply doesn’t stand fully due to flaw in experimental design.

6.    Limitations of the study should be explicitly called ‘limitations’, and serious flaw in experimental design should be, if not covered by inspecting more brightness magnitudes of the same or of other images, then at least by more thoroughly and concisely presented parameters which would cover this flaw. At the same time, further developments within previously presented experimental and methodological boundaries in this subject are strongly supported.

Author Response

Dear reviewer:

Aiming at questions your proposed for my manuscript, here are my explanations.

The question 1: Line 220 - Figure 3. missing
The question 2: Equation from line 175 - put in separate line: equation (4) equation (4) becomes (5) etc
The question 3: References&authors often not related to text: e.g. ref 206 line 77, ref 33-36 with line 203 etc.

1. The reply to these questions.

 We have modified these format problems in the manuscript.

 2. The question 4: Subjective judgement (mentioned in line 265, later in 265) must always be presented with and followed by construction and validation protocol.
The reply to fourth question.

Frankly, I did not quite understand the meaning of this question. I have added the relevant parameters of the listed algorithms in the line 293-302.

 3. The question 5: The question 5: Figures 7g) and 8 g) offer good representation of effects of proposed algorithm, mostly supported by values within objective criteria. However, presented example may seem highly selective or accidental, therefore concluding sentence in line 381 and 382: that the proposed method achieves significant denoising and detail enhancement effects on photon counting images obtained under illumination levels of 10-3 to 10-4 lux , simply stand fully due to flaw in experimental design.

The reply to fifth question.

I think there are part of reasons that presented example may seem selective or accidental, because Fig. 8(g) has more obvious denoising and enhanced effect than Fig. 7(g) even if Fig. 7(a) is collected in a worse illumination condition. The results are not corresponding to our expectation owing to worse illumination condition means greater difficulty in denoising, the biggest possibility is that Fig. 7(a) is got by scanning the Lena image, as we all know, it is a standard, digital image, The energy of each frequency band in the image is very rich: there are low frequency (smooth skin) and high frequency (feather on the hat). At the same time, it has better contrast and brightness information, which is very suitable for verifying various algorithms. The reason to choose 10-3 - 10-4 lux, we have explained it in lines 412-423.

The question 6: Limitations of the study should be explicitly called limitations , and serious flaw in experimental design should be, if not covered by inspecting more brightness magnitudes of the same or of other images, then at least by more thoroughly and concisely presented parameters which would cover this flaw. At the same time, further developments within previously presented experimental and methodological boundaries in this subject are strongly supported.

The reply to sixth question.

We added more parameters setting aiming at different illuminations level in the paper, which can help readers re-achieve relevant experiments.

Round  2

Reviewer 1 Report

Please see my detailed report. When responding to the reviewer's comments, it will be better to repeat each comment first and then present the response.

Author Response

The reply to the questions:

Question1: The objective evaluation results using PSNR,MSE, SSIM, etc. are not correct. The author responded by saying that they used Fig. 7(b) and Fig. 8(b) as references. This is also incorrect. Since 7(b) and 8(b) are histogram enhanced images, they should not be used as ground truth in the PSNR,SSIM,MSE, and IE calculations. For example, the image quality of Fig.8(b) is noisy. How can one use a poor quality image as the ground truth? What I mean is that one cannot use a low quality image as the ground truth to generate those PSNR, SSIM , etc. scores. The whole approach is flawed. Let me explain a little further. Suppose there is a denoising approach that generates a denoised image that is exactly the same as the HE image, then the PSNR score will be infinite. Does this mean this denoising approach is the best one as compared yo your proposed approach? Obviously, the answer is NO. In other words, no denoising will be considered as the best method. In my opinion, if the ground truth images are not available, one should not include those PSNR,MSE, SSIM, etc. in the results because they are meaningless.

The reply to the question1: Thanks for your advice. I have found the existing flaws in my manuscript and deleted the relevant contents of PSNR,MSE, SSIM indices, only preserving information entropy and two blind IQA indices to evaluate the quality of image.

Question2: I also have some concerns about PSNR,MSE, SSIM numbers in Tables 3 and 4.since the authors used Fig.7(b) and Fig.8(b) as references in generating those PSNR,MSE, SSIM values, then I expect the WA and WA-FICA should have the highest PSNR score because the WA and WA-FICA results look a lot closer to the HE image. Please double check your results.

The reply to the question2: We have deleted the PSNR,MSE, SSIM indices because it can not satisfy the conditions of usage, there is no distorted image as reference. This leads to huge errors of the results and not convincing.

Question3: The literature survey is not complete. I suggested one blind quality paper for remote sensing application and a relevant patent for denoising, but i did not see those in the revised paper. I suggested the authors should comment on those references and include them in the revised paper. This will make the references more complete.

The reply to the question3: I have researched and added the relevant literature you suggested in my manuscript and commented them.in literature[24] and [42].

Question4: On page13, the line number enter into Table 4. this should have been avoided

The reply to the question4: I have modified it

Question5: There are some typos like 2.19*10-3. Please check the manuscript more carefully. In L 182,198,etc., “Where” should be “where”.

The reply to the question5: Aiming at these questions, I have checked it carefully.

Question6: The definition of IE is not given in the paper. Please include an equation to show how it is generated

The reply to the question6: The equation and definition is given in manuscript

Reviewer 2 Report

Using full name of MPPC.

Author Response

thanks for your advice. I have used the full name of MPPC in the manuscript.

Reviewer 3 Report

Authors accepted suggestions and recommendations. Manuscript can be accepted in this present corrected form. 

However, standardization issue, i.e. construction and validation of evaluation criteria may not be understood only trough listing relevant parameters - it is a statistical precondition necessary for further generalizations and applications. 

Author Response

Thanks for your advice. I will consider standardization issue in the future research. when considering construction and validation of evaluation criteria, not only to list relevant parameters, but also to summarize universality and application from the global.